# Evaluation of Non-Invasive Sampling Techniques for the Molecular Surveillance of Equid Herpesviruses in Yearling Horses

**DOI:** 10.3390/v16071091

**Published:** 2024-07-07

**Authors:** Amjad Khan, Edward Olajide, Madeline Friedrich, Anna Holt, Lutz S. Goehring

**Affiliations:** 1Department of Veterinary Science, Martin-Gatton College of Agriculture, Food and the Environment, University of Kentucky, Lexington, KY 40506, USAl.goehring@uky.edu (L.S.G.); 2Department of Public Health & Nutrition, University of Haripur, Haripur 22600, Pakistan; 3College of Veterinary Medicine, Lincoln Memorial University, Harrogate, TN 37752-8245, USA

**Keywords:** gammaherpesvirus, PCR, air sampling

## Abstract

Background: Equid alphaherpesvirus 1 (EHV-1) is a highly contagious respiratory tract pathogen of horses, and infection may be followed by myeloencephalopathy or abortion. Surveillance and early detection have focused on PCR assays using less tolerated nasal swabs. Here, we assess non-invasive non-contact sampling techniques as surveillance tools in naturally equid gammaherpesvirus 2-shedding horses as surrogates for EHV-1. Methods: Horses were individually housed for 10 h periods on 2 consecutive days. Sampling included nasal swabs, nostril wipes, environmental swabs, droplet-catching devices, and air sampling. The latter was completed via two strategies: a combined air sample collected while going from horse to horse and a collective air sample collected at a stationary central point for 6 h. Samples were screened through quantitative PCR and digital PCR. Results: Nine horses on day 1 and 11 horses on day 2 were positive for EHV-1; overall, 90.9% of the nostril wipes, 81.8% of the environmental surfaces, and 90.9% of the droplet-catching devices were found to be positive. Quantitative analysis showed that the mean DNA copies detection per cm^2^ of nostril wipe sampled concentration (4.3 × 10^5^ per day) was significantly (*p* < 0.05) comparable to that of nasal swabs (3.6 × 10^5^ per day) followed by environmental swabs (4.3 × 10^5^ per day) and droplet catchers (3.5 × 10^3^ per day), respectively. Overall, 100% of the air samples collected were positive on both qPCR and dPCR. In individual air samples, a mean concentration of 1.0 × 10^4^ copies of DNA were detected in per m^3^ air sampled per day, while in the collective air samples, the mean concentration was 1.1 × 10^3^. Conclusions: Environmental samples look promising in replacing direct contact sampling. Environmental and air sampling could become efficient surveillance tools at equestrian events; however, it needs threshold calculations for minimum detection levels.

## 1. Introduction

Equid alphaherpesvirus 1 (EHV-1) is a global threat to equine health and welfare, as infection is associated with respiratory tract disease, detrimental spinal cord disease (EHM—equine herpesvirus-associated myeloencephalopathy), abortion, and neonatal death. Initial replication in the respiratory tract makes EHV-1 a highly contagious pathogen for other horses due to direct/indirect horizontal transmission including airborne spread [1]. EHM outbreaks are more common in boarding facilities; however, they have also been described in equine hospitals or during equestrian events [2,3]. Early detection of a shedding horse is key to outbreak prevention or mitigation. 

Currently, a shedding animal is diagnosed via swabs collected from nasal passages or nasopharynx and analyzed by (quantitative) polymerase chain reaction (PCR). Some horses will respond viciously to sampling procedures, becoming a danger to themselves and their surroundings, including humans. Alternative sampling procedures have been explored with good correlation to nasal swabs [4]. Yet, most alternatives still require an individual (horse) approach, which results in large sample sizes and contact with increased risk for fomite transmission. Here, we want to explore alternative methods of sample collection including air sampling. 

Herpesviruses of horses either belong to the subfamily’s *alpha* or *gammaherpesviridae*. [5]. EHV-1 is an alpha and EHV-2 is a gammaherpesvirus. Their replication characteristics in the upper respiratory tract epithelium and shedding pattern into the environment as smear or dispersed as respiratory tract droplets are believed to be very similar. EHV-2 is an extremely common pathogen in juvenile horses, while EHV-1 is not [6,7]. Furthermore, EHV-2 is usually shed for weeks up to months in most members of a closed herd. This is different from the group of alphaherpesviruses. Prolonged shedding is also a characteristic of human Epstein–Barr virus (cause of mononucleosis and a gammaherpesvirus) when compared to the human alphaherpesviridae members Varicella–Zoster or Herpes simplex virus. We felt equid gammaherpesvirus 2 (EHV-2) a safe choice to use as a surrogate for EHV-1 to study environmental surveillance and detection methods. 

The role of several transmission pathways of EHV-1 remains poorly understood. Growing evidence supports a combination of direct and indirect transmission [8]. Despite the importance of understanding the dynamics of airborne transmission, limited research has been conducted in this area. Considering this critical gap of knowledge in EHV transmission dynamics, this study sought to evaluate non-invasive sampling techniques as a molecular surveillance tool using a cohort of naturally EHV-2 shedding yearling horses. This study attempts to pave the way for a more reliable and feasible protocol for large-scale EHV-1 surveillance using non-invasive sampling.

## 2. Materials and Methods

### 2.1. Source of Animals 

This study was approved by the University of Kentucky Institutional Animal Care and Use Committee (2023-4251). A herd of 11 yearling horses naturally shedding EHV-2 were available at that time and kept in pasture as a group. These horses were previously used in an equine influenza virus (H3N8) experimental infection study (4 months earlier). Sampling was conducted on 2 consecutive days in a BSL-2 barn during July 2023. The facility is a closed concrete construction with a centralized ventilation system. The barn contains 12 individual 3.5 × 3.5 m stalls that open to a central barn aisle via sliding gates (Figure 1). Stalls have closed walls to neighboring stalls and a single window to the outside that remained closed during the experiment, and the sliding gates (1.2 m × 2.3 m) were made of vertical metal bars, which were the only communication to the shared airspace. 

Horses were transferred from pasture into the barn for 2 daytime periods of 9 h (07:30–16:30) on 2 consecutive days. Each animal occupied an individual and same (shavings bedded) stall during the 2 experimental days. Ad libitum hay was supplied inside the stall and was positioned directly in front of the sliding gate, and an automated waterer was located on the right side of the entrance in every stall. Both strategies were necessary to keep the horses’ heads closer to the gates.

### 2.2. Sampling Strategy

The same sample sets were collected on the 2 barn days. We collected 6 different samples: (i) nasal swabs; (ii) nasal ‘wipes’, (iii) surface swabs (inside a stall); (iv) droplet catching device (outside a stall); (v) air sample protocol 1 (all horses combined); (vi) air sample protocol 2 (stationary collection). For the nasal swabs, we used two 6″ (152 mm) long rayon-tipped swabs (Puritan^®^ Sterile Rayon Tipped Applicators, Guilford, ME, USA). For the nasal wipes, we used 5″ (127 mm) long rectangular (20 mm^2^) foam head swabs (skin surface area swabbed: 2.54 cm^2^) (VWR— Radnor, PA, USA). For the surface swabs, we used polyurethane sponges (80 mm^2^) with an 8″ long attached shaft (Whirl Pak Poly Probe 10 mL Hi Cap neutralizing broth, USA). On both days (occasions), the same pre-determined surfaces were sampled. For the droplet-catching devices, we used sterile (standard) petri dishes with a poly probe attached. The devices were placed in front of the gate at 0.5 m facing the stalls for a duration of 6 h on both days. For air sampling, we used a Coriolis™ Compact (Bertin, Montigny-le-Bretonneux, France) at maximum capacity of 50 L/min of air sampling. We sampled via 2 strategies: combined sampling for 4 min per horse moving from horse to horse (method 1) or a stationary protocol of continuous sampling for 6 h at a central point (method 2; position see Figure 1). Both sampling methods collected into device-specific dry collection cones.

### 2.3. Sampling Sequence

We collected nasal swabs and wipes at the beginning of the stabling period. Both surface samples were collected at the end of the stabling period before horses were released into pasture. Air samples were always collected at the device’s maximum collection rate (50 L/min) [4,9,10,11]. Individual air sampling (method 1) was completed going from stall to stall, and each stall was sampled for 4 min, sampling 200 L of air per horse (total: 2.2 m^3^), keeping the device close to each horse’s head and nose (distance: approx. 0.5–1 m). The collective air samples (method 2) were performed by placing the air sampler stationary at the center of the barn aisle one meter above the ground for 6 h, sampling a total of 18 m^3^ air. We collected a ‘control’ sample (4 min of sampling per empty stall) one day before the start of the experiment. The sequence of direct samples was nostril wipe first, which was followed by nasal (passage) swab. A wipe was collected from the ventral nostril area at or near the mucocutaneous junction. Then, two nasal swabs combined were inserted into the nasal passage on the same side. Swabs or wipes were immediately placed in individual 2 mL (empty) tubes. Then, a separate person followed with an individual animal air sampling (method 1). Then, we placed a droplet-catcher device in front of each gate of a stall accessible at 1.2 m height from the floor and 30 cm from the gate in the aisle. At the end of each experimental day, we collected all 11 droplet catchers first. Sponges were immersed into PBS (5 mL) in 50 mL tubes. Then, with a wetted surface swab, we gently rubbed over a 2 × 3 cm area of a horizontal bar of the sliding gate at the inside of each stall. Before animals were released into pasture for the night, we collected the collective-stationary sample (cone) and secured the sampler. 

### 2.4. Samples Processing

All samples were transported on ice to the laboratory; after a decontamination step at a separate sample-receiving area, samples entered the main laboratory. Once at the laboratory, 1 mL of PBS was added to each nasal swab in tubes inside a biosafety cabinet. Tubes were pulse vortexed for 90 s. Similarly, 5 mL of PBS was added to the tubes that contained the nostril wipes, droplet catchers, and environmental surface samples, which were then vortexed (90 s). Air sampler dry cones were rinsed with 5 mL of PBS, which was transferred into 15 mL tubes. All samples were frozen at −20 °C. 

### 2.5. DNA Extraction and Analysis

Samples were thawed in batches of 11 samples in a random fashion. Individual samples were vortexed for 10 s and centrifuged briefly. DNA from 200 uL of original sample was extracted using kits (DNeasy blood & Tissue Kit, Qiagen, Radnor, PA, USA) as per the manufacturer’s recommendations. Extracted nucleic acids were tested for the presence of EHV-2 using gB gene assays using a quantitative PCR (qPCR) system (Quant Studio 7, Applied Biosystems, Foster City, CA, USA) with standard thermal cyclic protocol [12]. Master Mix (TaqPath^TM^ qPCR Master Mix, ThermoFisher Scientific, Florence, KY, USA) and specific primers and probe (Table 1) mix (5.5 uL) for EHV-2 gB gene detection (both: ThermoFisher Scientific, Florence, KY, USA ThermoFisher) were mixed with 4.5 uL of extracted DNA through an epMotion setup (epMotion^®^ 5075, Hamburg, Germany). The 384 plates were sealed and centrifuged at 500× *g* for 2 min. The qPCR EHV-2 positive was reported qualitatively (presence or absence) and semi-quantitatively as cycle threshold (CT) values. Absolute quantification of the positive samples was completed through Applied Biosystems digital (d)PCR (Quant Studio Absolute Q Digital PCR, ThermoFisher Scientific Inc., Florence, KY, USA) using absolute Q DNA Digital PCR Master Mix™ (5x ThermoFisher Scientific, Florence, KY, USA) with the same EHV-2 gB primers and probe combination (Table 1).

### 2.6. Statistical Analysis

We conducted a statistical analysis using R Studio Version 2023.09.1. We began by measuring the normality of the data using the Shapiro–Wilk test. Having deviation from normality, we used a robust approach, continuing with a repeated-measures ANOVA to assess within-subject variability across different sample types. Despite the departure from normality, this approach allowed for a comprehensive evaluation of the impact of sample type on gB gene copy counts. To further validate our findings, we extended this analysis with a Kruskal–Wallis test, robust against abnormal data, which confirmed recorded variations in gene copy counts among sample types. 

Pairwise Wilcoxon rank-sum tests were then utilized for thorough comparisons to identify those in agreement with nasal swabs in gB gene copy detection. Furthermore, Bland–Altman plot analysis was conducted for a comparative analysis of EHV-2 detection in nasal swab samples versus the non-invasive samples. We also applied Fisher’s exact test to assess the percent agreement of a non-invasive sampling technique with nasal swabs. 

## 3. Results

Results of both days combined showed that all nostril wipes (100%), individual air samples (100%), and combined air samples (100%) were positive for EHV-2 genome copies via qPCR. Of all samples (both days), 90.9%, 81.8%, and 90.9%, respectively, were positive for nasal swabs, environmental swabs, and droplet catchers (Table 2). The highest mean gB gene copies detection per sample was in nasal swab samples, which was followed by droplet catchers, nostril wipes, environmental swabs, individual air, and combined air samples, respectively (Table 2). We recalculated for the differences in storage volume (1 mL vs. 5 mL). Estimating gB copies per sample, and the surface area that had been sampled, we detected a significantly higher concentration of EHV-2 virus nasal wipes per cm^2^ nares (skin) area sampled (Table 2) than all other samples collected. Individual air samples (method 1) performed better than method 2 for air sampling. All nasal swabs were tested for EHV-1 and EHV-4, and all were found to be negative. Therefore, we did not test the other samples for EHV-1 or EHV-4. 

To evaluate further relation and agreement between the non-invasive sampling techniques versus nasal swabs, we initiated our analysis by measuring the normality via a Shapiro–Wilk test, which showed a significant deviance from a normal distribution (W = 0.38449, *p* < 2.2 × 10^−16^). Given this, we opted for a robust approach. Regardless of the non-normality, we continued with repeated-measures ANOVA analysis. The ANOVA results specified a significant effect of sampling type on DNA copy counts (F = 3.04, *p* = 0.0080). However, our data did not fully meet the normality assumption; this approach allows for additional thorough evaluation. To further support our findings, we followed this analysis with a Kruskal–Wallis test, which is robust against non-normality and confirmed the observed differences in gB gene copy detection counts across sample types. Here, the results indicated that there is no significant difference (medians of each group) in gB gene copy detection among the various sample types (χ^2^ = 7.784, df = 6, *p* = 0.254). These findings were associated with our earlier observation of non-normality in the data and supported the employment of non-parametric tests. Further pairwise assessments were conducted to identify specific sample types that varied significantly from each other and detected the closest number of gB gene copies versus nasal swabs.

To perform pairwise comparisons between different sample types and to identify which one finds the most similar number of gene copies in comparison to nasal swabs, pairwise Wilcoxon rank-sum tests were used following a Kruskal–Wallis test. Non-invasive sampling types were further investigated for those closely matched nasal swabs for gB gene copy detection through dPCR. The *p*-values in Table 3 represent the level of agreement. A higher *p*-value indicates a stronger agreement, suggesting similar detection capabilities. Nostril wipes and droplet-catcher devices demonstrated good agreement to nasal swabs, indicating a good performance. Additionally, individual air and combined air samples also exhibited a high level of agreement with each other, suggesting similar detection outcomes between them (Figure 2).

Agreement between nasal swabs and other sample types was assessed (Figure 2) for individual samples in terms of gB gene copies detection via Bland–Altman plot [13]. In Figure 2 the red dotted line at 0 represents the mean difference level, indicating no systematic deviation between the methods. Droplet-catcher devices demonstrate the widest agreement range, indicating a higher variability of detection compared to other non-invasive methods. These results were further supported by analyzing the data through Fisher’s exact test to assess the detection capabilities of each method (Table 4).

## 4. Discussion

Considering the recent EHV-1 outbreaks reported at both equestrian events and boarding facilities [3,14,15,16,17,18], there is an increased need for the early diagnosis and detection of viral presence on premises that house susceptible animals. Currently, EHM outbreak prevention or mitigation relies on earliest case identification. The focus has been on identifying horses with fevers, symmetrical limb edema or sudden neurological gait anomalies However, clinical signs do not necessarily develop in each infected animal, while there is profound viral shedding into the environment [19]. 

Here, we used EHV-2 as a surrogate for advancing EHV-1 surveillance and early detection strategies [6]. We showed a good correlation between nasal swabs and alternatives. Notably, this is the first time that air sampling has been used for natural EHV shedding detection. We sampled single time point areas (nasal swabs and nares) and cumulative areas (over time). We distinguished between direct in-contact areas (surface areas inside a stall), areas of indirect transmission (droplet catcher), and air space [20]. 

Our main goals were twofold: we wanted to test horse-friendlier sampling alternatives other than nasal swabs along with sampling methods that do not require direct (hands-on) contact with a horse. In this experiment, nostril wipes were slightly superior over nasal swabs. The reason could have been either because the nasal wipe sampling area is larger (2.5 cm^2^ area) compared to the surface of two nasal swabs or because the sampled nostril area should be considered a cumulative collection area since respiratory tract secretions can accumulate as dried-in product, which can still contain detectable viral DNA. It is unclear how long viral DNA can be detected in wipes once nasal shedding ceases. It is probably no longer than 24–48 h considering epithelial tissue renewal; however, it could still be an important indicator for viral presence and should prompt the sampling of neighboring horses. Accumulation could have also been the reason for an observed increase in detected copy numbers from day 1 to day 2. These results are supported by previous reports [21,22]. 

With the environmental surface samples, we compared a collection surface with both options for direct as well as indirect transmission (area inside the stall) with an area outside the stall (droplet catcher, strictly indirect transmission). Significantly higher copy numbers were detected ‘inside’ compared to ‘outside’ of a stall. ‘Inside’ is more likely a combination of direct (smear) and indirect (droplets) accumulation. Furthermore, while the exposure time of both areas/devices remained the same, the distance from the source (horse) was also greater for the droplet catcher [23]. The sensitivity of surface swabs was low in comparison to nasal swabs. Results showed that four of the surface swabs were found negative on day 1; however, this also corresponds to individual low-shedding animals in their respective positive nasal swab samples. An additional explanation could be individual animal behavior and the varied ‘time spent’ near the gate area. 

It is reasonable to assume that EHV is both transmitted through droplet transmission (particles > 5 µm) and via aerosolized particles (<5 µm) [23]. For indirect transmission assessment at varying distances, we adopted two strategies: one to detect any airborne particles via droplet-catcher devices and via air sampling. However, it is important to realize that PCR detection will not represent the presence of infectious viruses. Positive results via qPCR could be for non- or no-more infectious virus particles of those attached to various particle types found inside a barn, including dust, dirt, pollen, skin debris (dandruff), and hair. These particles in a barn environment are often stirred up by either human and or horse movement and will become airborne as viral particles contributing to qPCR-positive results without necessarily indicating the presence of the infectious, viable virus. This highlights the need for further research to distinguish between infectious and non-infectious viral particles detected during indirect transmission to better understand EHV transmission dynamics.

The catcher devices with a surface area of 4.5 cm^2^ were found positive with significantly higher copy numbers (D1: 3.7 × 10^3^ and D2: 3.3 × 10^5^ copies per cm^2^ surface area of catchers) of EHV-2 than the (in-stall) surface samples on day 2 (Table 1). It was evident from our individual sample variation that horses shedding at higher concentrations reflected high detection levels of EHV-2 in droplet-catcher devices. We have enough evidence for the persistence of EHV in the environment [22,24]. The low detection in our study on day 1 and elevated detection level on day 2 could be attributed to several factors. It could be the fact that animals were getting used to the experiment, and they possibly stayed closer or more frequently to the gate area on day 2. It could also be due to random sneezing, coughing, or vocalizing episodes directly in the direction of droplet catchers. Materials could have had a different capture capacity, and it could be strictly by chance. However, the results show that there is a (short distance) transfer of viral genome and a good option for molecular detection without entering a horse’s stall [25].

Our second strategy was to investigate air sampling following technology gathered during the COVID-19 pandemic [8,9,10]. We either sampled a total of 2.2 m^3^ of air by going from horse to horse during 44 min of sampling time or by collecting 18 m^3^ of indoor air via a stationary position of the device over a runtime of 6 h. Both sampling strategies were successful in detecting either 9.0 × 10^3^ (Day1) and 1.1 × 10^4^ (Day2) genome copies per m^3^ air sampled or 1.1 × 10^3^ (Day1) and 1.2 × 10^3^ (Day2) DNA copies per m^3^ air sampled. Based on our results, we know (via nasal swabs) that all/most horses were shedding on both days at the time point of air sampling. An important difference is a constant distance (<1 m) between the shedding animal and the sampling device during the individual sampling strategy and a more variable distance (2–8 m; see Figure 1) with method 2. Distance is essential, as viruses will move further if attached to small particulates or aerosols rather than attached to droplets or other large particles, while droplets can carry more viral particles. However, this is a proof-of-concept study. In it, we were able to detect virus via both sampling techniques. At this point we recognize that the sampling concept needs refinement. For example, there is a need to determine a successful sampling strategy if only one horse is shedding virus. We also need to define the duration of sampling (runtime of the equipment), the maximum distance to the sampling source, and what may be optimal stable conditions for sampling (e.g., ventilation characteristics during stabling periods, other environmental factors).

While these are promising results for sampling strategies ‘without touching a horse’, both environmental and air samples should be collected to allow for maximum sensitivity and surveillance. However, we collected a total of 200 L of air by running the air sampler for 4 min in front of each stall. In this air sample collected via method 1, we successfully detected 5.9 × 10^3^ copies per cubic meter of air. Similarly, an air sample via method 2 was collected by putting a stationary air sampler for 6 h at the center of the aisle at approximately 2–2.5 m away from the bars of the closest set of 4 stalls. Through this method, we detected significantly lower concentrations than with individual air sampling. Reasons could be the distance of placing the air sampler away from the source shedding point, as distance affects the spread of the virus and its dilution in air [20]. These results show airborne spread EHV-2 (genome) in agreement with previous studies [26], reporting aerosolization and the indirect spread of both EHV-1 and 4 [27,28]. Interestingly, in this study, the air sampler via method 1 qPCR detection was in good agreement with nasal swabs, while catcher devices had better agreement with nasal swabs in terms of detection with a sensitivity of 95%. For direct detection nostril wipes, and for indirect detection air sampling, methods 1 and 2 were found promising and convenient. But in our case, 90.9% of horses were shedding both days. In addition to the data provided by this pilot study, we have plans to pursue threshold determination as well as sensitivity and specificity analyses.

## 5. Conclusions

Environmental samples look promising in replacing direct contact samples with the animals. Air sampling could become an elegant surveillance technique for respiratory pathogens at equestrian events; however, it will need threshold calculations to be able to detect a single case and a further optimization of sampling techniques.

## Figures and Tables

**Figure 1 viruses-16-01091-f001:**
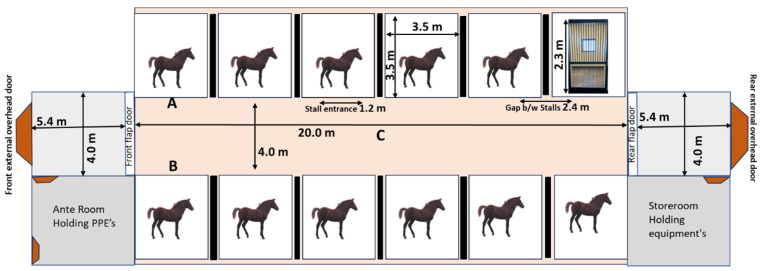
Lay out of the BSL-2 barn and experimental model design. Location of droplet-catcher device in front of each gate (A, B). Position of air sampler during stationary protocol for 6 h runtime (C).

**Figure 2 viruses-16-01091-f002:**
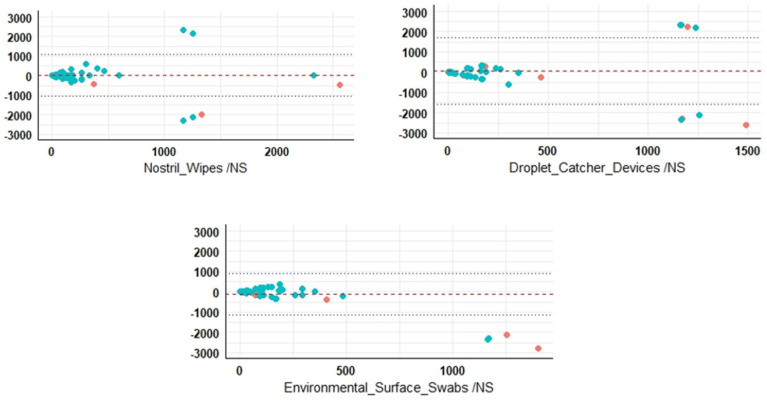
Comparative analysis of EHV −2 gB gene detection: nasal swab versus non-invasive sampling techniques. This Bland−Altman plot illustrates the agreement between nasal swab measurements and those obtained from other sample types. The red dashed line represents the average difference, showing that, on average, the methods provide similar results. The blue dotted lines indicate the range of agreement (95% limits of agreement), providing context for data spread. The Y-axis represents the differences between measurements, with 0 denoting no difference. Positive and negative values indicate over- and underestimation, respectively.

**Table 1 viruses-16-01091-t001:** EHV-2 specific primer and probe sequence used for PCR amplification in qPCR and dPCR.

Primer	Sequence
EHV2 gB-Fw EHV2 gB-Rev EHV2 gB-Probe	CGCAGAGGATGGAGACTTYTTACA CATGACCGTGGGGGTTCAA 6FAM-CTGCCCGCCGCCTACTGGATC-BHQ1

**Table 2 viruses-16-01091-t002:** EHV-2 detection results via qPCR in all types of samples. Absolute quantification of qPCR positive samples through digital PCR expressed as gB gene copies per µL of purified DNA.

Variables	Nasal Swab	Nostril Wipes	^1^ Environmental Sponges	^2^ Droplet Catchers	^3^ Individual Air	^4^ Combine Air
No of positive (n) samples in total Samples (N), (n/N)	D1: 9/11 D2: 11/11	D1:11/11 D2:11/11	D1:7/11 D2: 11/11	D1:9/11 D2:11/11	D1:2/2 D2:2/2	D1:1/1 D2: 1/1
Approximate Surface area and air volume sampled	2 swabs (combined)	2.54 cm^2^	6.45 cm^2^	4.5 cm^2^	2.2 m^3^	18 m^3^
Approximate calculated Mean gB gene copies detected (dPCR)	D1: 3.5 × 10^5^ D2: 3.6 × 10^5^ (per 2 swabs)	D1: 3.1 × 10^5^ D2: 5.6 × 10^5^ (per cm^2^)	D1: 5.1 × 10^4^ D2: 3.5 × 10^4^ (per cm^2^)	D1: 3.7 × 10^3^ D2: 3.3 × 10^5^ (per cm^2^)	D1: 9.0 × 10^3^ D2: 1.1 × 10^4^ (per m^3^ air)	D1: 1.1 × 10^3^ D2: 1.2 × 10^3^ (per m^3^ air)

^1^ Sampled at the end of the experiment day 1 and 2 (8 h horses stayed in stalls). ^2^ Droplet catchers were installed for 6 h. ^3^ Individual stalls were sampled by placing the air sampler for 4 min in front of each stall at a speed of 50 L/minute. ^4^ Combined air samples were collected by placing the air sampler running in the center of the aisle for 6 h.

**Table 3 viruses-16-01091-t003:** Pairwise Wilcoxon rank-sum test for detection agreement between sample types.

Sampling Types	Nasal Swabs	Nostril Wipes	Environmental Swabs	Droplets Catchers	Individual Air	Combined Air
Nasal Swabs	NA	0.71	0.15	0.62	0.07	0.29
Nostril Wipes	0.71	NA	0.05	0.49	0.04	0.18
Environmental Swabs	0.15	0.05	NA	0.54	0.72	0.67
Droplets Catcher	0.62	0.49	0.54	NA	0.62	0.71
Individual Air	0.07	0.04	0.72	0.61	NA	1.00
Combine Air	0.29	0.18	0.67	0.71	1.00	NA

The *p*-values presented in the table indicate the level of agreement between the sample types. A *p*-value closer to 1 signifies a higher level of agreement, suggesting similar detection outcomes between the compared sample types.

**Table 4 viruses-16-01091-t004:** Sensitivity-based agreement between nasal swabs and non-invasive sampling techniques.

Sampling Type	Positive Percentage	Agreement with NS *	Fisher Exact Sig **
Nostril Wipes	100%	100%	1.00
Environmental Swab	81.8%	80.0%	0.66
Droplet Catchers	90.9%	95.0%	0.83
Individual Air	100%	100%	1.00
Combine Air	100%	100%	1.00

* Nasal swab (NS) samples were used as standard, and the rest of the sample types were compared to it. ** Fisher exact significance value >0.05 shows no significant difference between the two sampling methods.

## Data Availability

Data will be made available on request.

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
