# Peer review of "Evaluation of Non-Invasive Sampling Techniques for the Molecular Surveillance of Equid Herpesviruses in Yearling Horses"

_viruses, 2024, doi:10.3390/v16071091_

Round 1
Reviewer 1 Report
Comments and Suggestions for Authors
This is an interesting study the authors should be commended
Sadly the manuscript is lacking in many areas and lets down the study somewhat including Grammar and explanation to readers of the methods and logic
I have made some suggestions but suggest an overhaul of the manuscript with greater clearer explanation to the reader

/
Author Response
From Amjad Khan,
Corresponding Author:
Manuscript ID: viruses-3051807,
Title: Evaluating non-invasive sampling techniques for the molecular surveillance of Equid herpesvirus (EHV) in naturally shedding yearling horses
Journal: [Viruses] MDPI.
Dated: June 22, 2024.
Subject: Response to Reviewers Comments.
To,
The Editor Office Viruses,
Thanks for your email of June 15, 2024, forwarding us the reviewers’ comments. We have addressed all the comments in the revised manuscript and each point has replied accordingly. Point by point reply and the details of the revisions and responses to the referees’ comments are given below.
Reviewer#1 comments and reply
General Comment:
Evaluating non-invasive sampling techniques for the molecular surveillance of Equid herpesvirus in naturally shedding yearling horses
This is a sophisticated piece of work well done several areas are vague and need qualifying and more explanation for the readers
Response: Thanks. We have tried to improve the revised version thoroughly by having it reviewed by native English speaker.
Comment: Consider rewording title to make clearer eg.
Evaluation of non-invasive sampling techniques for the molecular surveillance of Equid herpesviruses in yearling horses
Response: We have accepted this consideration and have revised the title in our revised version of the Manuscript.
Comment: And thoroughly explain your choice to sample ehv 2 when youre looking to EHV 1 –
detail all aspects please. Requires further explanation and qualification to make clear difference btw alpha and gamma
Response: As per reviewer suggestion, we have added explanation in the introduction section highlighted regarding, why we used EHV-2 as a surrogate in more details in revised version online#58 to 69.
Comment: I would remove the term gold standard for nasal swabs (what about N/P swabs!) rather this is the traditional/standard form of sampling. In this study according to your objective’s gold standard would be comparing EHV-1 nasal swab detection with your noninvasive methods.
Response: We have accepted the suggestion and the term gold standard have been replaced with Nasal swabs in the revised version of the manuscript. The changes have been highlighted in the revised version.
Comment: Explain detection comparison for EHV-2 gamma vs EHV-1 alpha provide more explanation around this please (Methods)
Response: We have added few lines to explain the detection comparison briefly for EHV-2 gamma versus EHV-1 alpha revised version online#58 to 69.
Comment: (Methods) Details of horses? number history/background.
Response: As per reviewer’s suggestion details of the horses, total number and their history is added in the revised manuscript online#81 to 83.
Comment: Results: Nine horses on day 1, and 11 horses on day 2 were positive to EHV-1 following nostril wipes, 81.8% of environmental surface, and 90.9% of droplet catching devices. Quantitative analysis 29 showed that mean DNA copies detection per cm2 of nostril wipe sampled concentration (4.3x105 per 30 day) was significantly (P) comparable to
Response: we have made these changes accordingly in the revised version at line#29-31, these changes are highlighted.
Comment: Discussion
Many areas of poor grammar need to check some examples highlighted below. Considering the recent EHV-1 outbreaks reported both at equestrian events and 245 boarding facilities [3, 13-15], enforce the need of early diagnosis and detection of viral 246 presence on premises housing susceptible animals. Currently, EHM outbreak prevention 247 or mitigation relies on earliest case identification Line 247
The focus has been on identifying horses with neurological deficits like including ataxia ver, symmetrical urinary incontinence and limb edema or sudden neurological gait 249 anomalies. However, clinical signs do not necessarily develop in each infected, or clinical signs develop during.
Response: We have taken help of two native English speakers to review and correct the poor grammar issues. The revised manuscript is now hopefully free of English grammatic errors. All the changes made are highlighted in the revised version.

Reviewer 2 Report
Comments and Suggestions for Authors
The authors investigated non-invasive sampling techniques using environmental swabs and droplet catchers as surveillance tools in naturally equi gammaherpesviru 2 shedding horses as surrogate for equid alphaherpesvirus 1 (EHV-1), which is a pathogen causing respiratory infection, abortion, neonatal death, and equine herpesvirus myeloencephalopathy in horses. Effective control measures are required to prevent the spread and transmission of EHV-1 in horses. Invasive sampling including nasal swabs and nostril wipes has been used for diagnosis and the molecular surveillance of EHV infections. The experiments in the present manuscript were scientifically planned and examined. The manuscript seems to be written logically. The non-invasive sampling techniques present in the manuscript might be valuable as a herd diagnosis. However, the conclusion that environmental samples look promising in replacing invasive sampling is over evaluation, because it is unable to identify which horse is infected using environmental samples. The non-invasive sampling techniques present in the manuscript might be useful for additional methods for the molecular surveillance of EHV-1 infection in horses.
L57-59: The authors used equid gammaherpesvirus 2 (EHV-2) as a surrogate for EHV-1 in the present study. The reason is that EHV-2 is a common and relatively benign pathogen in juvenile horses with a reference 5. The authors should show much more why EHV-2 can be a surrogate for EHV-1 such as the same transmission route of EHV-2 and EHV-1 and others.
L111-112: The authors cited references 4 to 9 to explain air sampling. The references 5 and 6 did not describe anything about air sampling. Citation should be [4, 7-9].
L178-179:The authors described that positive rates were 81.18% for nasal swabs, 72.7% for environmental swabs, and 59.1% for droplet catchers. However, data in Table 2 were 9/11 (81.8%) of D1, 11/11 (100%) of D2 and 20/22 (90.9%) of both days for nasal swabs, 7/11 (63.6%) of D1, 11/11 (100%) of D2 and 18/22 (81.18%) of both days for environmental sponges, and 9/11 (81.8%) of D1, 11/11 of D2 and 20/22 (90%) of both days for droplet catchers. How did the authors calculate the positive rates shown in Line178? The authors should add the number of positive samples to the total samples in Line178 to clarify the positive rates and match with data in Table 2.
Line 254: The authors cited reference 4 in their explanation about EHV-2 as a surrogate. However, there is no description about EHV-2 in reference 4. It should be reference 5 as cited in Line 59. Line 285: The authors described "particles > 5 µM". Why did the authors use "M" which should mean "mol per liter" as a chemical unit usually Do they use "M" for "m" (meter)? The authors have to use international units correctly.
Author Response
Reviewer#2 comments and reply
General Comment: The authors investigated non-invasive sampling techniques using environmental swabs and droplet catchers as surveillance tools in naturally equi gammaherpesviru 2 shedding horses as surrogate for equid alphaherpesvirus 1 (EHV-1), which is a pathogen causing respiratory infection, abortion, neonatal death, and equine herpesvirus myeloencephalopathy in horses. Effective control measures are required to prevent the spread and transmission of EHV-1 in horses. Invasive sampling including nasal swabs and nostril wipes has been used for diagnosis and the molecular surveillance of EHV infections. The experiments in the present manuscript were scientifically planned and examined. The manuscript seems to be written logically. The non-invasive sampling techniques present in the manuscript might be valuable as a herd diagnosis. However, the conclusion that environmental samples look promising in replacing invasive sampling is over evaluation, because it is unable to identify which horse is infected using environmental samples. The non-invasive sampling techniques present in the manuscript might be useful for additional methods for the molecular surveillance of EHV-1 infection in horses.
Comment: L57-59: The authors used equid gammaherpesvirus 2 (EHV-2) as a surrogate for EHV-1 in the present study. The reason is that EHV-2 is a common and relatively benign pathogen in juvenile horses with a reference 5. The authors should show much more why EHV-2 can be a surrogate for EHV-1 such as the same transmission route of EHV-2 and EHV-1 and others.
Response: We have added more details regarding the use of EHV-2 as a surrogate for EHV-1 with references in the revised version on the line#58 to 69.
Comment: L111-112: The authors cited references 4 to 9 to explain air sampling. The references 5 and 6 did not describe anything about air sampling. Citation should be [4, 7-9].
Response: The citation order has been corrected in the revised version and highlighted at Line#113.
Comment: L178-179: The authors described those positive rates were 81.18% for nasal swabs, 72.7% for environmental swabs, and 59.1% for droplet catchers. However, data in Table 2 were 9/11 (81.8%) of D1, 11/11 (100%) of D2 and 20/22 (90.9%) of both days for nasal swabs, 7/11 (63.6%) of D1, 11/11 (100%) of D2 and 18/22 (81.18%) of both days for environmental sponges, and 9/11 (81.8%) of D1, 11/11 of D2 and 20/22 (90%) of both days for droplet catchers. How did the authors calculate the positive rates shown in Line178? The authors should add the number of positive samples to the total samples in Line178 to clarify the positive rates and match with data in Table 2.
Response: The positivity of the samples was calculated in term of frequency (percentages) by dividing total positive out of total number of the samples. Initially the samples were tested on qPCR and then re-tested on digital PCR (dPCR) which is more sensitive specifically for such samples where we have low copy numbers per microliter of samples. Therefore, we reported the percentages calculated from data of qPCR results but as per reviewer’s suggestion we have revised these numbers to make them uniform across the article based on digital PCR results to reflect same numbers according to table 2. The changes are highlighted in grey color.
Comment: Line 254: The authors cited reference 4 in their explanation about EHV-2 as a surrogate. However, there is no description about EHV-2 in reference 4. It should be reference 5 as cited in Line 59.
Response: The reference number cited had been corrected in the revised version.
Comment: Line 285: The authors described "particles > 5 µM". Why did the authors use "M" which should mean "mol per liter" as a chemical unit usually Do they use "M" for "m" (meter)? The authors have to use international units correctly.
Response: Microns, also known as micrometers (represented as µm) are a length of measurement equal to one millionth of a meter. (1,000µm is equal to 1mm.). The official symbol for the micron or micrometer is μm, sometimes simplified as um. A micron is defined as one-millionth of a meter, a little more than one twenty-five thousandth of an inch.
We have followed this unit from a published literature given here “Pusterla N., Mapes S. Evaluation of an Air Tester for the Sampling of Aerosolised Equine Herpesvirus Type 1. Vet. Rec. 2008; 163:306–308. doi: 10.1136/vr.163.10.306”.
The same united has been used in a recent publication which is cited as “Dayaram A, Seeber PA, Greenwood AD. Environmental Detection and Potential Transmission of Equine Herpesviruses. Pathogens. 2021 Apr 1;10(4):423. doi: 10.3390/pathogens10040423. PMID: 33916280; PMCID: PMC8066653”.

Round 2
Reviewer 1 Report
Comments and Suggestions for Authors
sadly still some grammatical error s please re read
examples
Line 52 delete to
Resent to is poor grammar again ;/
Line 2527delete need to for
Need for
Comments on the Quality of English Language
see above disappointing that two english speakers still submit errors in grammar /
Author Response
Comment #1: Line 52 delete to
Response: We have changed the sentence to "Some horses will respond viciously to sampling procedures, becoming a danger to themselves and their surroundings including humans" on line # 52 to 53.
Comment#2: Line 2527delete need to for.......Need for.
Response: We have revised the sentences which had suggested grammar issues on Line# 331 to 337.
and Line#354 to 356.
Note to the Editor: We were confused by the comments of reviewer 2 regarding ‘need to for’. We had all sections which contain ‘need to’ reviewed or revised by a senior British native. Please advise whether we correctly understood the comments of reviewer 2.

Reviewer 2 Report
Comments and Suggestions for Authors
L294: The authors misunderstood my comment. The authors misused "M" (upper case M) as a unit for length of the particle. "5 µM" should be "5 µm". "M" is a unit for concentration (mol/L). The authors should read comments carefully.
Author Response
Comment: L294: The authors misunderstood my comment. The authors misused "M" (upper case M) as a unit for length of the particle. "5 µM" should be "5 µm". "M" is a unit for concentration (mol/L). The authors should read comments carefully.
Response: We are grateful to the reviewer for pointing out the error again. We have corrected the Upper case M to m on L294 in the revised Manuscript.

Round 3
Reviewer 2 Report
Comments and Suggestions for Authors
The authors improved the manuscript well. The manuscript will provide us with new insights on EHV-1 infection research.